# Same Brain, Different Look?—The Impact of Scanner, Sequence and Preprocessing on Diffusion Imaging Outcome Parameters

**DOI:** 10.3390/jcm10214987

**Published:** 2021-10-27

**Authors:** Ronja Thieleking, Rui Zhang, Maria Paerisch, Kerstin Wirkner, Alfred Anwander, Frauke Beyer, Arno Villringer, A. Veronica Witte

**Affiliations:** 1Department of Neurology, Max Planck Institute for Human Cognitive and Brain Sciences, 04103 Leipzig, Germany; imrui.zhang@gmail.com (R.Z.); paerisch@posteo.de (M.P.); fbeyer@cbs.mpg.de (F.B.); villringer@cbs.mpg.de (A.V.); 2Institute for Medical Informatics, Statistics and Epidemiology (IMISE), University of Leipzig, 04107 Leipzig, Germany; kwirkner@life.uni-leipzig.de; 3Leipzig Research Center for Civilization Diseases (LIFE), University of Leipzig, 04103 Leipzig, Germany; 4Department of Neuropsychology, Max Planck Institute for Human Cognitive and Brain Sciences, 04103 Leipzig, Germany; anwander@cbs.mpg.de; 5Day Clinic for Cognitive Neurology, University of Leipzig Medical Center—Leipzig University, 04103 Leipzig, Germany; 6Berlin School of Mind and Brain, Humboldt-Universität zu Berlin, 10117 Berlin, Germany

**Keywords:** diffusion magnetic resonance imaging, white matter, fractional anisotropy, multi-centre, reproducibility, imaging artefacts, ageing

## Abstract

In clinical diagnostics and longitudinal studies, the reproducibility of MRI assessments is of high importance in order to detect pathological changes, but developments in MRI hard- and software often outrun extended periods of data acquisition and analysis. This could potentially introduce artefactual changes or mask pathological alterations. However, if and how changes of MRI hardware, scanning protocols or preprocessing software affect complex neuroimaging outcomes from, e.g., diffusion weighted imaging (DWI) remains largely understudied. We therefore compared DWI outcomes and artefact severity of 121 healthy participants (age range 19–54 years) who underwent two matched DWI protocols (Siemens product and Center for Magnetic Resonance Research sequence) at two sites (Siemens 3T Magnetom Verio and Skyra^fit^). After different preprocessing steps, fractional anisotropy (FA) and mean diffusivity (MD) maps, obtained by tensor fitting, were processed with tract-based spatial statistics (TBSS). Inter-scanner and inter-sequence variability of skeletonised FA values reached up to 5% and differed largely in magnitude and direction across the brain. Skeletonised MD values differed up to 14% between scanners. We here demonstrate that DTI outcome measures strongly depend on imaging site and software, and that these biases vary between brain regions. These regionally inhomogeneous biases may exceed and considerably confound physiological effects such as ageing, highlighting the need to harmonise data acquisition and analysis. Future studies thus need to implement novel strategies to augment neuroimaging data reliability and replicability.

## 1. Introduction

Diffusion-weighted imaging (DWI) is a widely established, powerful and non-invasive in vivo magnetic resonance imaging (MRI) technique used in human clinical and non-clinical applications [1,2]. DWI measures water diffusion in biological tissue, which is hindered and restricted, for example, by fibre bundles, cell membranes and other cell structures in the brain. This renders DWI a valuable tool to acquire in vivo information of brain properties at a microscopic scale [3]. For example, diffusion tensor imaging (DTI) uses the diffusion of water molecules to determine the static anatomy of the brain (not influenced by brain function), yielding different tensor-based measures such as fractional anisotropy (FA), which is the degree of directionality of water diffusion within brain tissue, and mean diffusivity (MD), which describes the molecular diffusion rate of water within brain tissue. Thereof, axonal fibre tract coherence and structural connectivity as well as microstructural properties of the white and grey matter can be estimated [3,4,5]. DWI/DTI is both noise-sensitive and prone to imaging artefacts due to, e.g., eddy currents, susceptibility-induced distortions, Nyquist ghosting or physiologically related factors (e.g., cardiac pulsation and subject motion) [6]. Therefore, technicians and scientists put continuously high effort into developing improvements for hardware and software [6]. Since the introduction of DTI in the mid-1990s [3], MRI techniques developed towards higher magnetic fields, stronger gradients and more sensitive detectors; thereby the signal-to-noise ratio and spatial resolution of MR images in general and of DTI in particular improved. This has led to a better understanding of structural connectivity [7,8] and to the discovery of changes in microstructure due to, e.g., ageing and neurodegenerative diseases [9,10,11,12] as well as experience-dependent plasticity [13,14,15,16]. With increasing availability of research-oriented MRI assessments on a larger scale in the last decades, such as in (multi-centre) longitudinal clinical trials and epidemiological cohorts reaching hundreds and even thousands of measurements (e.g., Human Connectome Project [17], UK Biobank [18], German National Cohort MRI Study [19]), however, developments in MRI hardware and software start to outrun the extended periods of data acquisition and analysis of these studies. Thus, DTI studies often experience changes during data acquisition like improvements of scanning protocols, the development of new sequences or minor to major hardware changes such as scanner upgrades. Nevertheless, it has been more and more acknowledged that not only obvious MRI artefacts per se but also subtle changes in sequence parameters and scanner hardware can systematically affect outcome measures [20,21,22,23,24]. Therefore, ensuring the comparability of DTI-derived outcome measures across scanners—but also within scanner and across sequences— is of uttermost importance.

Previous attempts to estimate the inter-site reproducibility of DTI-derived measures reported divergent results, e.g., with regard to how large a potential inter-site difference would be [25,26,27,28,29,30,31,32,33,34,35,36,37,38,39,40,41]. Outcomes from a common approach to assess human brain white matter microstructure (i.e., tract-based spatial statistics (TBSS) [42] of FA maps) comprise for example coefficients of variation (CoV) ranging from 1.0% [38] to 4.1% [28] to 14.4% [37] for inter-site reproducibility. However, certain methodological shortcomings limit the validity of these studies. For example, if only a phantom was scanned on several imaging sites [31,35,37], it is hard to interpret how differences in magnetic field homogeneity or gradient fields would affect the results for a human brain or body. In addition, in studies analysing human brain DW images, subject number was either low (*n* = 1 [32], *n* = 1 [28], *n* = 1 [25], *n* = 1 [[26], *n* = 2 [36], *n* = 3 [40], *n* = 5 [41]) or different individuals underwent DWI at the imaging sites that were compared [29,30,40,43]. In the latter case, results do not reflect scanner intrinsic variances but are biased by differences in the individual microstructure. Additionally, the time passed between scans was up to one year [40] or even up to 22 months [27] which makes differences in brain microstructure difficult to attribute solely to differences between MR scanners (and not to physiological changes). Other limitations are the choice of a rather narrow age range (67–84 years old [43]; 8–19 years old [29]; 50–58 years old [33]) as well as the inclusion of a neurologically non-healthy subject (relapsing-remitting multiple sclerosis [26]; mild cognitive impairment and Alzheimer’s Disease [43]) thus results cannot be generalised. Collecting DW images with a variety of manufacturers [25,28,30,31,40], sequence parameters [30,40] and field strengths [31] introduces even more uncertainties. Due to technical advances, a significant number of studies are affected by scanner upgrades. Nevertheless, to our best knowledge, there has only been one investigation of DTI metrics on scanner platform effects (pre- and post-upgrade), and only for 3T General Electric MRI scanners with a limited sample size [34]. To summarize, these methodological considerations highlight the need for more comprehensive inter-site comparability studies.

Besides acquisition-related differences in DTI-derived outcome measures that possibly originate from physically inherent differences between scanners and protocols, these differences may in addition fortify due to imaging artefacts. While several a posteriori correction methods have been implemented in commonly used preprocessing software to mitigate such imaging artefacts [44,45,46], one of the most ubiquitous artefacts, the Gibbs ringing (GR), received less attention. Only in recent years, attempts addressing the removal of this artefact have been published [47,48,49,50,51,52]. GR appears due to a k-space truncation along finite image sampling and presents as signal oscillations at sharp intensity transitions leading to physically implausible signals (PIS) and erroneous FA values (e.g., FA > 1), thus potentially wrong interpretations of the underlying microstructure. Physically implausible FA values show as lines of bright voxels at tissue boundaries in the FA images. Even though this adverse impact of GR on DTI-derived metrics has been recognised for almost three decades [47,49,52,53,54,55], until recently, no state-of-the-art preprocessing pipeline such as FSL [44,45] had a GR artefact removal tool included yet. Whether such preprocessing would indeed lessen acquisition-related differences in DTI-derived outcome measures has not been evaluated yet. Nevertheless, throughout the evaluation of a novel GR artefact removal tool—the “Kellner Method” [48]—it has already been integrated in the MRtrix3.0 [6] preprocessing pipeline.

We aimed to systematically determine the effects of different scanner versions and preprocessing pipelines on DTI-derived outcomes using a large sample size. Specifically, we compared a Siemens 3T Magnetom Verio with its upgraded version Siemens 3T Magnetom Skyra and evaluated the “Kellner Method” (a Gibbs ringing artefact removal tool) [48] in comparison to the standard low-pass window filtering technique available on the Siemens scanner, noise reduction (MRtrix3.0) [6,56] and a pipeline without correction. We chose a within-subject design, short time gap between scans, high number of subjects and matched scanning protocols at both imaging sites—scanning protocols are publicly available at https://osf.io/vnuqp/ (accessed on 5 July 2021). The main research questions of the current study comprise:What is the reproducibility of DTI-derived measures across-scanners (with differing upgrade versions) using high-resolution diffusion-weighted MRI on two 3T high-field scanner systems?What is the intra-site but across-DWI-sequences comparison of DTI outcome measures from two sequences with matched protocols?What is the impact of different preprocessing tools on measurement reproducibility (image denoising, GR artefact reduction, default low-pass window filtering)?What are the conclusions to be drawn from the abovementioned results in relation to physiological effects (such as ageing) on white matter FA?

## 2. Methodology

### 2.1. Participants

121 healthy participants (60 female, age range 19 to 54 years, 29.9 ± 8.2 years old) were invited to undergo two head MRI acquisitions lasting about 75 min each. Exclusion criteria were MRI contraindications such as implanted medical device, metal fragment in the body, or claustrophobia as well as pregnancy, neurological or psychiatric conditions and centrally effective medication. Participants were scanned at two different imaging sites. Five participants did not return for the second appointment (rescanning), resulting in a total of 116 participants for analyses. The interval between individual scanning sessions ranged from 2 to 139 days, and all scans were acquired within a 5-month-period. The study was approved by the Research Ethics Committee of the University of Leipzig and was conducted in accordance with the Declaration of Helsinki. All subjects gave written informed consent and received reimbursement for participation.

### 2.2. MR image Acquisition

DWI scans were performed on two common 3T Siemens MRI scanners, namely Magnetom Verio (Syngo MR B17) and Magnetom Skyra^fit^ (Syngo MR E11) (Siemens Healthineers GmbH, Erlangen, Germany). These two scanner versions are often linked through an upgrade from Verio to Skyra^fit^. The upgrade would include the replacement of all hard- and software parts except for the main magnet and the gradient coil. All 121 participants were at first scanned on Verio at the Day Clinic of Cognitive Neurology at the University of Leipzig and then on Skyra at the Max Planck Institute for Human Cognitive and Brain Sciences, Leipzig. A counterbalanced order of scanners could unfortunately not be realised for organisational reasons. To assure a reproducible image acquisition, the brains of all participants were carefully positioned in the centre of the gradient system with a standardised head positioning procedure in order to minimise distortions and b-value variations caused by gradients non-linearities.

On both scanners, we used a 32-channel head coil and two double spin-echo encoding sequences which lasted 16 min 8 s. Throughout this work, we will refer to the two different protocols with “MPIL” (Siemens product sequence) and “CMRR” (developed by Moeller et al. [57] at the Center for Magnetic Resonance Research, University of Minnesota) ([TR]/[TE]: 13800/100 ms, 72 slices, 60 diffusion directions (b = 1000 s/mm^2^), 7 non-diffusion-weighted volumes (b = 0 s/mm^2^), EPI-factor: 128 (resolution 128 × 128), FoV: 220 × 220 × 123 mm^3^, voxel size: (1.7 mm)^3^, Phase Partial Fourier: 6/8 (MPIL) and 7/8 (CMRR)). Parallel imaging was performed in both protocols with a generalised auto-calibrating partially parallel acquisition (GRAPPA), reconstruction algorithm and an acceleration factor of 2. The CMRR protocol was identical at both scanners, only the MPIL protocol had to be slightly adjusted at the 3T Magnetom Skyra to [TR]: 14,400 ms due to the duty cycle limitations of the Skyra system relative to Verio. Nevertheless, this difference should not affect diffusion imaging as the white and grey matter spin systems ought to be relaxed to equilibrium after any [TR] >10 s and therefore, the MPIL protocol can also be viewed as identical at both scanners. The CMRR protocol was run with Siemens product low-pass window filtering option of the raw data to reduce high frequency imaging artefacts such as GR whereas the MPIL protocol was run without it and DW images were in addition retrospectively reconstructed on the scanner console with the Siemens product low-pass window filter. Thereby, we were able to assess the quality of this low-pass window filter. 

For anatomical reference, a high-resolution 3D structural image was acquired for all participants at each scanning site. Therefore, we used a magnetisation-prepared 180 degrees radio-frequency pulses and rapid gradient-echo (MPRAGE) sequence with the following parameters: [TR]/[TI]/[TE]: 2300/900/2.98 ms, 176 slices, flip angle: 9°, FoV: 256 × 240 × 176 mm^3^, voxel size: (1 mm)^3^, GRAPPA-factor 2. Scanning protocols are available at https://osf.io/vnuqp/ (accessed on 5 July 2021).

### 2.3. Image Processing

Raw image data was exported as DICOMs and transformed to NIfTI format [58]. During this step b-values and b-vectors were extracted. Further, we applied a denoising tool from MRtrix3.0-rc1 [6] in order to reduce signal fluctuations originating in thermal noise. This preprocessing step is supposed not to target Gibbs ringing artefacts. It is recommended to denoise the images before approaching the removal of the Gibbs ringing artefact as the denoising tool (MRrtix3.0 (“DWI Denoising—MRtrix 3.0 Documentation”)) [56] detects and removes noise characteristics which would be altered by any additional preprocessing step. To evaluate different preprocessing techniques for diffusion-weighted images, we compared four different preprocessing pipelines as shown in Figure 1. The main focus was to compare the Siemens low-pass window filtering and the “Kellner Method” [48] which address the removal of the Gibbs ringing artefact. The other two comprised neither noise nor Gibbs ringing correction nor only noise correction.

Further preprocessing steps included the segmentation and removal of non-brain tissue with bet (Brain Extraction Tool) embedded in FSL [59]. With the FSL software eddy [46], we implemented the replacement of slices showing signal drop-outs due to subject head motion. Next, we applied motion correction, and rigid body registration to each participant’s own skull-stripped and AC-PC-reoriented T1-weighted image in one step together with the interpolation of the target isotropic resolution of 1 mm with a tool developed in-house called Lipsia [60]. In the final preprocessing step, the diffusion tensor was modelled and metrics like FA and MD were estimated at each voxel using Lipsia again. To account for motion-attributed changes in the DTI parameters, we estimated frame-to-frame head motion by calculating the frame-wise displacement (FD, in mm) across volumes [61] using the 6-parameter motion output generated from eddy [46]. This mean FD was used as a covariate to correct for head motion in statistical analysis [62].

### 2.4. Quality Assessment

Initial visual Quality Assessment (QA) was conducted to ensure data fidelity. No individuals had to be excluded due to structural abnormalities. We conducted further quality checks after every main preprocessing step. To be exact, we assessed the overall quality of the diffusion data by inspecting the signal-to-noise-ratio maps of the b0 images and the contrast-to-noise-ratio maps of the b1000 images as well as the residuals (difference between the observation and Gaussian process predictions) with FSL’s eddy for imaging artefacts. Before statistical analyses, we reviewed the registration of all FA maps to the common template (FMRIB58_FA in MNI space from FSL) [59] to confirm the precission of this step which is crucial for our region of interest approach.

### 2.5. Region of Interest Approach

In order to assess differences in mean FA and mean MD values introduced by imaging site, sequence and composition of the preprocessing pipeline not only on the whole brain level but also for different regions, we extracted mean FA and MD values from fibre tracts that were defined in line with [38]—namely the splenium of the corpus callosum (SCC) (selected manually), left superior longitudinal fascicle (LSLF) and left uncinate fascicle (LUF) (ROIs highlighted in Appendix A). Before masking the mean FA and MD skeleton to obtain mean values for the above listed ROIs, we non-linearly warped the ROIs from the JHU-ICBM atlas (1 mm) to the FMRIB58_FA MNI space (both from FSL).

### 2.6. Statistical Analysis

To test whether differences in scanners (Verio vs. Skyra), sequences (MPIL vs. CMRR) or preprocessing tools (unfiltered vs. denoised vs. Siemens low-pass window filtering vs. unringing by Kellner method) affect DTI-derived outcome measures, we analysed whole-brain voxel-wise and ROI-based mean FA and MD values within the white matter skeleton (see Figure 1). 

Through tract based spatial statistics (TBSS) [42], we obtained FA and MD maps of the white matter skeleton for each subject in each condition. Briefly, all FA maps were co-registered using affine and non-linear transformations to standard space and the individual local maximal FA values were projected onto the standard FA skeleton to match individual’s anatomy. The threshold for these standardised white matter fibre tract maps was set at 0.2. In order to obtain the MD skeleton maps for each subject, we applied the non-linear warps and skeleton projection from the FA processing to the MD data. The FA and MD skeleton maps were lastly fed into voxel-wise analysis of FA and MD for statistical comparison using the randomise tool by FSL version 5.0.1. We used 1000/2000 permutations and threshold-free cluster enhancement as test statistic. With this tool, we conducted voxel-based paired two-sample *t*-tests on white matter skeletons of each subject to detect locations which differed significantly (*p*-value (FWE) < 0.05). Voxel-wise analysis was conducted on a whole brain level and the FD estimates across volumes were included as a covariate of no interest. In addition, we extracted and compared the average skeletonised FA and MD values in the three different ROIs (Appendix A) to compare broader regional variations.

On the whole brain level, we further compared mean FA and MD values of the WM skeleton with Bayesian linear modelling and Bayesian paired two-sample *t*-tests with the “BayesFactor” package included in R. A quick guide to Bayesian statistics: the Bayes Factor is defined as the following ratio: $BF=\frac{likelihood\_of\_data\_given\_H1}{likelihood\_of\_data\_given\_H0}$. Conventionally, the alternative hypothesis H1 (“there are one or more effects”) is more likely if BF>3 and the null hypothesis H0 (“data is random noise”) is accepted if $BF<\frac{1}{3}$. A Bayes Factor between $\frac{1}{3}$ and 3 suggests that the data are not informative regarding which hypotheses should be accepted. Bayesian statistics were additionally applied on FA and MD values of the WM skeleton in selected ROIs (see “Region of Interest Approach” above). 

#### 2.6.1. Inter-Scanner Variability

For the analysis of the scanner comparison (Verio vs. Skyra), we focused on the DW images recorded with the CMRR sequence from 115 subjects in order to guarantee high statistical power and on the “state-of-the-art” preprocessing pipeline including denoising. Data from the CMRR sequence of one of the 116 subjects scanned at both imaging sites were corrupted and the subject had to be excluded. We applied both, the whole brain and ROI-based FA and MD approach using TBSS. We further calculated the differences of the mean FA and MD values in percentages on a per subject basis: $\frac{mean\;FA\;value\;\left(Skyra\right)\;-\;mean\;FA\;value\;\left(Verio\right)}{mean\;FA\;value\;\left(Verio\right)}\;\times \;100$ or $\frac{mean\;MD\;value\;\left(Skyra\right)\;-\;mean\;MD\;value\;\left(Verio\right)}{mean\;MD\;value\;\left(Verio\right)}\;\times \;100$.

#### 2.6.2. Inter-Sequence Variability

Regarding the sequence comparison (MPIL vs. CMRR), we excluded datasets recorded which were acquired with different reconstruction parameters, leaving us with 51 subjects for the MPIL sequence and their matched scans from the CMRR sequence (Verio *n* = 23, Skyra *n* = 28). Deviations from the measurement protocol comprised the missing retrospective reconstruction with the Siemens product window filtering (Verio, MPIL sequence, *n* = 93) and the application of data interpolation during reconstruction (Skyra, MPIL sequence, *n* = 88). Voxel-wise statistical comparison of the sequences was then conducted with TBSS’ randomise tool on the FA and MD skeleton maps as described above.

#### 2.6.3. Gibbs Ringing (GR) Artefact

We visually assessed the reduction of GR artefacts by the different preprocessing pipelines (MPIL sequence, Verio *n* = 23, Skyra *n* = 28). In order to quantify the GR artefact, we extracted the number of voxels with implausible fractional anisotropy values (FA > 1) which are introduced by GR. Those voxels were clearly affected by GR and the amount of those voxels provided a conservative estimation to analyse if this number differs between scanners and preprocessing approaches using Bayesian statistics.

#### 2.6.4. Motion Effects

To ensure comparability of studies conducted at different scanners, we also looked into possible differences in subject head motion quantified as frame-wise displacement (FD, in mm) between scanners (with CMRR sequence, *n* = 115). Thereto, we fed mean FD values into Bayesian linear modelling. We further investigated if motion effects can be attenuated by certain preprocessing approaches, using Bayesian linear modelling and post-hoc paired two-sample *t*-tests. 

#### 2.6.5. Age Effect

We investigated age as a biological phenotype of interest and evaluated size differences of the negative effect of age on voxel-wise and whole brain mean FA and MD (CMRR sequence, *n* = 115). To this end, we performed additional analysis of data from the LIFE Adult Study (*n* = 1255) [63,64]. Based on this cross-sectional data, the negative age effect could be estimated to a decrease in mean FA of the WM skeleton of 0.14% per year. In order to simulate the case of data collection at different imaging sites, we compared not only the age effect on the dataset from Verio with the one from Skyra but also with a dataset consisting of randomly chosen DW images from Verio and Skyra (1:1 ratio, *n* = 50 from each scanner). 

#### 2.6.6. Harmonisation Attempt

In line with Pohl et al. [40], we calculated the ratio between the mean FA value of the whole brain’s WM skeleton from Skyra and Verio in order to possibly harmonise FA values across scanners. The ratio would serve as a correction factor (cf) to harmonise data before statistical analyses (meanFA(Verio) = cf × meanFA(Skyra)). Adding to the whole brain analysis, we also looked into the ratios for the selected ROIs.

### 2.7. Coefficient of Variance

Previous studies on DTI test–retest replicability commonly reported the coefficient of variation (CoV) as statistical measure [25,30,37,38,39]. The CoV, defined as the ratio of the measurement’s standard deviation σ divided by the mean μ and multiplied by 100 ($CoV\;=\;\frac{\sigma }{\mu }\;\times \;100$), served as an estimate of data dispersion expressed as relative percentage independent of the absolute measurement values. For the assessment of the inter-scanner variability, we calculated the CoV of WM skeleton mean difference FA and MD values (|Verio − Skyra| in %, single subject, voxel-based) after preprocessing with denoising on a whole brain level and in selected ROIs. Regarding the GR artefact and motion effects, we report respectively the CoV of the number of voxels with FA > 1 and of the mean FD values.

## 3. Results

### 3.1. Inter-Scanner Variability

We observed a significant difference between 3T Verio and Skyra scanners after preprocessing with denoising in whole brain white matter skeleton mean FA values (Figure 2 and Figure 3; CMRR sequence; Bayesian linear modelling, *n* = 115: BF[mean FA value ~ scanner] = 33.9) with a CoV of about 7.1%; this means that the alternative hypothesis H1 is 33 times more likely than the null hypothesis. On the whole brain level, Skyra delivered slightly higher mean FA values (see Table 1). However, scanner differences in mean FA value were not consistent across white matter tracts (Figure 2, Table 1). Central fibre tracts with high FA values such as the splenium of the corpus callosum (SCC) delivered significant differences between the mean FA values of the two scanners (Bayesian linear modelling, *n* = 115: BF[mean FA value ~ scanner] = 1.1 × 10^32^, CoV 30.7%) with Verio showing much higher values. More lateral such as the left uncinate fascicle (LUF) did not show significant differences between scanners (BF = 1.1) but a CoV of about 31.5%. However, in fibre tracts with mean FA values of the same scale (FA ≈ 0.5) but of a longer range such as the left superior longitudinal fascicle (LSLF), scanner differences were significant (BF = 3.3 × 10^12^) with Skyra showing higher values and a CoV around 11.1%. Differences of the mean FA values in percentages $\frac{mean\;FA\;value\;\left(Skyra\right)\;-\;mean\;FA\;value\;\left(Verio\right)}{mean\;FA\;value\;\left(Verio\right)}\;\times \;100$: whole brain: ~1%, SCC: ~−5.2%, LUF: ~1%, LSLF: ~1.1%. In addition to the analysis of scanner differences in the FA skeleton for the different brain regions, we also tested for scanner differences in MD values (Figure 4 and Figure 5). With TBSS, we found a clear whole brain difference with Skyra showing higher MD values than Verio (Figure 4; CMRR sequence; preprocessed with denoising). This clear direction of the scanner difference is supported by the whole brain mean MD value comparison with Bayesian linear modelling (Figure 5, Table 2; *n* = 115: BF[mean MD value ~ scanner] = 1.4 × 10^8^) with a CoV of about 11.1%. Central fibre tracts such as the splenium of the corpus callosum (SCC) and lateral fibre tracts such as the left uncinate fascicle (LUF) exhibit the same pattern (Skyra showing higher MD values than Verio) with differences in magnitude (SCC: BF[mean MD value ~ scanner] = 1.3 × 10^50^, CoV = 27.4%; LUF: BF[mean MD value ~ scanner] = 1.4 × 10^4^, CoV = 28.3%). Only in longer fibre tracts comprising many different and crossing fibre orientations, scanner differences are not evident (BF[mean MD value ~ scanner] = 0.16, CoV = 13.9%). Differences of the mean MD values in percentages $\frac{mean\;MD\;value\;\left(Skyra\right)\;-\;mean\;MD\;value\;\left(Verio\right)}{mean\;MD\;value\;\left(Verio\right)}\;\times \;100$: whole brain ~2%, SCC: ~14%, LUF: ~3%, LSLF:~−0.2%.

### 3.2. Inter-Sequence Variability

Investigating the effect of different imaging sequences run at the same scanner (with harmonised protocol parameters, MPIL vs. CMRR, Verio *n* = 23, Skyra *n* = 28), TBSS (TFCE and motion corrected, *p* < 0.05) detected that regional mean FA values differed significantly in several WM tracts dependent on scanner. Sequence differences were more pronounced in FA maps from Verio: CMRR showed higher FA values in central brain-areas, mainly in the CC, whereas MPIL showed higher FA values in cortical tracts in the left hemisphere (Figure 6a,b)). Data from Skyra though showed no WM tracts in which CMRR delivered higher FA values than MPIL but MPIL indicated higher FA values in both hemispheres, cortically and sub-cortically (Figure 6c,d)). Regarding the comparison of the sequences based on MD maps, patterns were less pronounced: the CMRR sequence appeared to deliver higher MD values in both hemispheres cortically and sub-cortically (Figure 7a,c)) whereas the MPIL sequence shows higher MD values cortically and rather frontally and in the left hemisphere at Verio (Figure 7b)) but more occipitally and in the right hemisphere at Skyra (Figure 7d)).

### 3.3. GR Artefact in DW Images 

The qualitative visual data control of the MPIL sequence (Skyra) revealed that the GR artefact appeared very strong in the unfiltered b0 (T2-weighted) images. After preprocessing, different levels of GR reduction were visually detected (Figure 8). Specifically, while denoising did not reduce GR artefacts, the unringing tool by Kellner et al. [48] seemed to clearly reduce the GR artefact. The low-pass window filtering by Siemens introduced a global blurring but the oscillations starting from the bright cortico-spinal-fluid (CSF) surrounding the brain were still visible.

In line with visual assessment, quantitative analyses indicated that different preprocessing pipelines (Skyra, MPIL sequence) differ significantly in their efficiency of reducing the amount of implausibly high FA values (Bayesian linear modelling: BF[#(voxels with FA > 1) ~ preprocessing pipeline] > 4.7 × 10^10^). The unringing tool reduced the amount of implausible FA values significantly (paired Bayesian *t*-test: BF[unfiltered ~ Kellner Method] > 8 × 10^9^, CoV = 42.3%) whereas the quantity of implausible FA values did not change consistently after any other preprocessing tool (BF[unfiltered ~ denoising] = 0.7, CoV = 104.7%), BF[unfiltered ~ Siemens low-pass window filtering] = 13, CoV = 78.1%) (Figure 9). Further, there were no significant differences in the amount of physically implausible FA values between Verio and Skyra (Bayesian linear modelling, full/null model comparison: $BF\left[\frac{\left(voxels\;with\;FA\;>\;1\right)∼scanner\;\times \;preprocessing\;pipeline}{\left(voxels\;with\;FA\;>\;1\right)∼preprocessing\;pipeline}\right]$= 0.05 ± 1.09%). 

### 3.4. Motion Effects

Comparing head motion quantified as mean frame-wise displacement (FD) values (in mm) between scanners (CMRR sequence, *n* = 115), it became evident that the estimated motion effects differed significantly (Bayesian linear modelling, full/null model comparison: $BF\left[\frac{mean\;FD\;value∼scanner\;\times \;preprocessing\;pipeline}{mean\;FD\;value∼preprocessing\;pipeline}\right]$> 4.6 × 10^70^ ± 1.96%). Regarding the significant differences between levels of preprocessing (Bayesian linear modelling, full/null model comparison: $BF\left[\frac{mean\;FD\;value∼scanner\;\times \;preprocessing\;pipeline}{mean\;FD\;value∼scanner}\right]$> 2.4 × 10^11^ ± 1.93%) (Appendix A), estimated motion effects could be attenuated significantly by applying denoising and unringing (see BF of post-hoc paired Bayesian *t*-tests in Table 3). Mean FD values are shown in Table 4. However, scanner differences remained significant even after further preprocessing (paired Bayesian *t*-test, *n* = 115, all BF > 10^18^).

### 3.5. Physiological Effects of Interest

The age effect on whole brain WM skeleton mean FA equalled approximately −0.06% per year (CMRR sequence, after unringing). The effect was estimated with linear modelling (mean FA/MD value ~ age) and comparable between scanners (estimate ± std. error): mean FA: Verio: −3.203 × 10^−4^ ± 1.237 × 10^−4^ and Skyra: −2.957 × 10^−4^ ± 1.347 × 10^−4^ (Figure 10); mean MD: Verio: −2.420 × 10^−7^ ± 1.654 × 10^−7^ and Skyra: −2.584 × 10^−7^ ± 1.722 × 10^−7^. Bayesian linear modelling confirmed the significance of the negative age effect on the FA skeleton in a full/null model comparison: $BF\left[\frac{mean\;FA\;value∼age\;\times \;scanner}{mean\;FA\;value∼scanner}\right]$= 5.49 ± 1.39%. but a potential age effect on mean MD value could not be confirmed with Bayesian linear modelling: $BF\left[\frac{mean\;MD\;value∼age\;\times \;scanner}{mean\;MD\;value∼scanner}\right]$= 0.23 ± 1.04%.

The negative age effect can be further depicted by TBSS on the FA skeleton and is observable in Verio and Skyra. By extracting *t*-values of the *t*-maps from TBSS, we could confirm that—in line with the absolute FA value approach—the age effect was slightly stronger in Verio (mean *t*-value = 0.517) than in Skyra (mean *t*-value = 0.453) (CMRR sequence, after unringing; Appendix A, images above, *t*-values averaged over all voxels). In the case of studies conducted on different scanners (simulated by randomly selected subjects from Skyra and Verio), the age effect size was of intermediate magnitude (Figure 10; Appendix A, image below).

### 3.6. Harmonisation Attempt

As suggested by Pohl et al. [40], we calculated a whole brain correction factor in order to potentially harmonise the data and obtained $\frac{mean\;FA\left(Verio\right)}{mean\;FA\left(Skyra\right)}$ = *cf = 0.9892* (after denoising), which was comparable between preprocessing pipelines (Bayesian linear modelling, *n* = 115: BF[mean FA ratio ~ preprocessing pipeline] = 0.6; Figure 11). Calculations of the correction factors for the selected ROIs (SCC, LSLF, LUF) however yielded correction factors which differed significantly (after denoising/after unringing with Kellner Method) depending on ROI: SCC = 1.055/1.0575, LSLF = 0.9885/0.9903, LUF = 0.9884/0.9881 (Bayesian linear modelling, *n* = 115: $BF\left[\frac{mean\;FA\;ratio∼ROI\;\times \;preprocessing\;pipeline}{mean\;FA\;ratio∼preprocessing\;pipeline}\right]$>> 2 × 10^196^ ± 1.15%.; Figure 11). The different levels of preprocessing did not play a major role here (Bayesian linear modelling, *n* = 115: $BF\left[\frac{mean\;FA\left(Skyra\right)\;-\;mean\;FA\left(Verio\right)}{mean\;FA\left(Verio\right)}\right]$< 7 × 10^−6^ ± 1.52%).

## 4. Discussion

Using a large sample size of healthy adults that underwent repeated MRI scanning at 3 Tesla with state-of-the-art acquisition and (pre)processing pipelines, we here report systematic global and regional differences in common DTI outcome measures between different scanners, sequences and pipelines. More specifically, we observed relative mean skeletonised FA value differences between scanners of up to 5% across brain regions and relative mean skeletonised MD value differences between scanner of up to 14%, which may well exceed potential effects of ageing (estimated to reach about 0.14%) or effects of disease on these measures. In addition, we found that the unringing tool from Kellner et al. [48] reduced Gibbs ringing artefacts satisfactorily as opposed to other preprocessing approaches without unringing. Head motion quantified as mean frame-wise displacement (FD) values were consistently lower in the scanning sessions at Skyra compared to Verio, and motion-related artefacts were additionally reduced after preprocessing by denoising or unringing.

### 4.1. Regional FA and MD Variability Due to Different Scanners and Sequence Parameters

Scanner differences in DTI outcome measures after “state of the art” preprocessing of DWI data with denoising range from about 1% globally to about 5% locally on the FA skeleton and from about 2% globally to about 14% locally on the MD skeleton. Our findings of these considerable variations across distinct ROIs of the WM skeleton’s mean FA and MD values have to our knowledge not yet been reported, and suggest further investigation including a careful evaluation of recent attempts to harmonise multi-centre DWI data (as suggested by Pohl et al. [40]): variations included not only large differences in magnitude but differed for mean FA also in direction (Skyra vs. Verio): mean FA: whole brain: ~+1%, SCC: ~−5.2%, LUF: ~+1%, LSLF: ~+1.1%; mean MD: whole brain ~+2%, SCC: ~+14%, LUF: ~+3%, LSLF: ~−0.2%. We further calculated the coefficient of variance (CoV) of WM skeleton mean difference FA and MD values (|Verio—Skyra| in %, single subject, voxel-based) after preprocessing with denoising on a whole brain level and in selected ROIs. This inter-scanner CoV for the mean difference FA ranged from ~7% globally to ~30% locally and is thereby locally much higher than inter-scanner CoVs from previous studies (1.0% [38] to 4.1% [28] to 14.4% [37]); similarly, the inter-scanner CoV for the mean difference MD ranged from ~11% globally to ~28% locally, also much higher than the according inter-vendor Siemens CoV from [25] of 4.4%. Our CoVs compared to past studies show that the mean FA and MD values might not be as robust to inter-scanner variations as previously assumed. 

Pohl and colleagues [40] harmonised scanner differences with correction factors for whole brain mean FA values regardless of variation across brain regions. This would lead to either fortified or attenuated local effects and is therefore not suitable for clinical diagnostics or studies focusing on regional effects. 

TBSS on the FA skeleton presented a pattern with higher FA values for Verio data in more superficial white matter and higher FA values for Skyra data in rather deep WM areas. We observed a bias similar to this pattern when comparing anatomical MR images from Verio and Skyra [65], namely Skyra exhibiting higher cortical thickness and larger GM volumes in medial frontal and central regions and Verio showing higher cortical thickness in lateral and occipital regions. This pattern could be caused by scaling differences between scanners that were observed on the anatomical images and could affect diffusion image processing due to registration on the individual anatomical images. TBSS on the MD skeleton showed that values from Skyra are almost uniformly higher throughout the whole brain. This might be due to slight miscalibrations of diffusion gradients between scanners. Yet, local differences on FA skeletons might be caused by differences in gradient non-linearities or other hardware or software differences.

These findings emphasise that a retrospective correction for scanning at different imaging sites is hardly possible. Therefore “imaging site” should always be considered as a covariate in statistical analyses. Possible sources for such large differences between scanners could lie in hardware (e.g., radio frequency transmission, receiver coil sensitivity or signal processing elements) or software (reconstruction algorithms, data processing) differences.

We also showed that DWI data from sequences with harmonised but not identical parameters collected at the same scanner present region-dependent differences in TBSS, which is in line with earlier studies suggesting a strong sensitivity of DWI and its outcome measures to sequence parameters [66]. The only difference between the CMRR and MPIL protocol identifiable with the scanner software was the amount of k-space reconstruction (partial Fourier). For EPI sequences, a large k-space coverage is necessary to reduce the EPI readout time and therefore increase the image quality, and thus 6/8 for MPIL and 7/8 for CMRR of k-space lines were acquired. Nevertheless, we cannot exclude that this sampling difference could cause slight image quality differences due to different k-space coverage. Yet, those are expected to be global differences in resolution, e.g., more blurriness for lower coverage, but not regionally specific effects. 

Of note, a negative influence of age on whole brain WM coherence (represented by skeletonised mean FA values) as physiological effect of interest is much smaller—0.06% reduction per year in this relatively young sample and 0.14% reduction per year in the additionally analysed older cohort of the LIFE Adult Study—than the differences introduced by multi-site (and also partly by multi-sequence) data collection.

Despite the relatively young cohort (29.9 ± 8.2 years old), we confirmed the negative effect of ageing on WM coherence (estimated cross-sectionally by FA) in analyses within-scanners. However, when pooling data from the two imaging sites, the physiological effect of age detected with TBSS was not extinguished but attenuated and changed in regional extent. We failed to detect an effect of ageing on the WM skeleton MD values possibly because FA may be a more specific measures of age-related changes in WM. Nevertheless, we conclude that pooling datasets from different imaging sites might fail to detect small effect sizes and/or may deliver regionally inconsistent patterns of the effect of interest if during analysis it is not accounted for the different imaging sites. This is especially crucial in clinical diagnostics if patients are scanned at different imaging sites. In this case, pathological changes could be attenuated or masked and therefore be missed. 

As we did not assess intra-scanner variability, by repeating the same imaging protocol on the same scanner, the observed differences might be partly due to intra-scanner variability. Yet, a previous study showed high reliability of intra-scanner DTI metrics which was similar to intra-session differences and mainly influenced by the applied preprocessing steps [67]. Therefore, our finding of systematic differences between scanners is likely to be driven mainly by inter-scanner variability, largely independent of intra-scanner variability. Nevertheless, future studies should incorporate a test/retest intra-scanner acquisition as to quantify the contributions of the different sources of variability.

To account for a spatial heterogeneity of scanner differences, Fortin and colleagues [29] suggested ComBat as a tool to harmonise FA and MD maps. ComBat is a batch effect correction tool used in genomics [68] which aims to remove site effects from DTI maps and seems to preserve biological phenotype such as age. Yet, the locally largely differing CoVs as well as their divergence from the whole brain CoV indicate that the extent of scanner differences is not consistent across regions and subjects, rendering retrospective correction difficult. Future analyses need to test if applying ComBat in multi-site DWI effectively reduces between-scanner variance. 

Taken together, our finding of severe regional differences in skeletonised FA and MD values between scanners and sequences strongly argue to keep imaging parameters stable if possible and to remain with data collection at one imaging site, or to increase sample sizes dramatically in multi-site studies to adjust for the reduction of statistical power. Introducing quality control protocols and phantom scans to characterize scanner systems in single- and multi-site studies can also help to increase the reliability of diffusion weighted measurements (as suggested by Fedeli et al. [69]). In the clinical context, we recommend to rescan a patient at the same MRI machine and to implement standardized quality controls.

### 4.2. Gibbs Ringing and Motion Artefacts

Regarding attempts to reduce common artefacts such as the Gibbs ringing (GR), we compared three different preprocessing approaches plus data without additional filtering and demonstrated that visually and quantitatively the unringing tool from Kellner et al. [48] reduced the GR artefact most efficiently. To the best of our knowledge, quantitative assessment of the GR has so far not yet been established with an easy, ready-to-apply method which is why we introduced the number of voxels with an implausible FA value (FA > 1) as an approximation of the amount of GR. We confirmed by visual inspection that the implausible FA values in the selected voxels (FA > 1) were caused by GR and not by other artefacts. GR can of course affect FA values without causing them to exceed 1, especially in areas with lower FA values, so that the amount of implausible FA values cannot be seen as an absolute measure of GR but rather as a conservative estimation of the number of voxels clearly affected by GR. Even though other measurement noise could affect FA to exceed 1 by, e.g., causing negative eigenvalues [70], other preprocessing tools such as the Siemens low-pass window filtering supposed to address this noise or the denoising tool from MRtrix did not attenuate the GR visually or the amount of FA > 1 quantitatively. Additionally, eddy current and vibration artefacts could lead to systematic patterns of artificially high FA values with a particular spatial pattern. In our experiment, eddy currents were successfully compensated by the twice refocused pulse-sequence, and we did not observe vibration artefacts in any of the measurements. We therefore conclude that FA > 1 is an appropriate lower approximation of GR in our data. We suggest inclusion of the unringing tool from Kellner et al. [48] in DW preprocessing in order to increase data quality and to possibly mitigate differences between data from different scanners before pooling them in a multi-site study. Promising future steps towards automatic GR artefact detection and reduction besides the Kellner tool might be the application of convolutional neural networks as suggested and experimentally verified by Zhang et al. [50], Zhao et al. [51] and Muckley et al. [52].

Regarding head motion, mean FD values were estimated consistently lower in the scanning sessions at Skyra. Even though, all participants underwent their second scan at Skyra, most participants are very MRI-experienced and therefore the chronology of the scanning sessions unlikely explains the considerable attenuation in head motion. While speculative, we suppose DWI is that demanding for the hardware such as the gradient coils that the wearing off during the years of use (Verio in use since 2008) compared to Skyra (upgraded in 2016) might have an effect on the increased estimated motion effects in Verio. The scanners might show a slight difference in the non-compensated eddy currents or a scanner drift which might result in a difference in the estimated head motion parameters by the FSL eddy tool. This tool estimates the eddy currents and head motion at the same time and both estimations are not independent. FD values could be reduced by including denoising or unringing in the preprocessing pipeline. This reduction in apparent head motion can be explained by an improved image quality introduced through these filtering techniques and therefore improved motion estimation. Nevertheless, differences in head motion were used as covariate in statistical analyses and did not influence scanner, sequence or pipeline differences significantly. 

### 4.3. Limitations and Strengths 

Our study includes two scanner systems of the same manufacturer and two diffusion-weighted sequences—a main limitation therefore is that it does not reflect the whole range of most commonly used MRI systems in clinics and research neither all of the most commonly implemented DWI sequences. However, considering that the two very similar scanner systems and two harmonised sequences exhibit considerable differences in DTI outcome measures, it can be speculated that more differing scanners and sequences exhibit more substantial differences. 

More detailed limitations include that we did not correct for gradient non-linearities which could similarly affect diffusion tensor metrics as the apparent diffusion coefficient as shown by Fedeli et al. [69] and Tan et al. [71] and thereby account for some of the differences we found on the mean FA and MD skeletons. However, the small non-linearity in the gradients of the used clinical MR systems, the relatively small field-of-view (only the brain) and the comparably low b-value (b = 1000 s/mm²) minimize the effect of gradient non-linearity on image distortions and b-value variation compared to other studies where such corrections are needed (e.g., Human Connectome Project [17]. To further control for a correct application of the diffusion gradients, the positioning of the participants followed a standardised protocol to position the brain in the isocentre of the gradient coil which presents the smallest non-liberalities. Additionally, in a parallel comparison of the anatomical images from Verio and Skyra, gradient non-linearity correction—conducted with the gradunwarp implementation [https://github.com/Washington-University/gradunwarp (accessed on 5 October 2021)] in Python 2.7.)—did not substantially reduce the detected differences [65]. This is why we estimate that gradient non-linearities might only explain a small share of the differences in the WM skeletons between the scanners. Nevertheless, we did not check and correct for gradient non-linearities in this study, and therefore cannot exclude small residual effects.

Concerning the quantification of the GR artefact, the linear least squares (LLS) method employed by Lipsia to estimate the diffusion tensor in our preprocessing pipeline, also used in widely established DWI preprocessing software such as FSL’s dtifit, comes with the negative eigenvalue problem. Negative eigenvalues can be caused by measurement noise and lead to FA values larger than 1. This is, for example, the case for voxels presenting the GR artefact where the signal of the diffusion weighted images is locally increased and might be higher than the non-diffusion weighted (b0) signal in the same voxel. The linear tensor fit leads to physically implausible negative eigenvalues (and therefore FA > 1) in those voxels. Koay et al. [70] showed in simulations that the constrained non-linear least squares (CNLS) method is, in terms of mean squared error for estimating trace and FA, the most effective method for correcting negative eigenvalues. Studies focussing on FA and areas with high anisotropy such as the corpus callosum should therefore reconsider the approach to estimate the diffusion tensor in order to ensure data quality. In our case, measurement noise might have been different in areas with high anisotropy leading to inflated differences in the FA skeleton between scanners and sequences, especially in the splenium of the corpus callosum (SCC). We also recommend consideration of employing the CNLS method to calculate tensor-based DWI metrics as suggested by Koay and colleagues [70] in order to minimize the effect of data heteroscedascity.

Additionally, as scanner order could not be randomised in this project due to scheduling issues, all participants underwent the first MRI at Verio and the second at Skyra which may have led to effects of scanning order we could not account for in our analysis. Further, we did not include a retest measurement on the same scanner to discriminate between within- and between-scanner effects. As shown by Fedeli and colleagues [72] in a large multi-centre phantom study, DWI metrics (in this case ADC values) and their spatial uniformity can differ significantly across ROIs at varying distances from iso-centre. Preceding phantom scans or the re-scan of a participant or patient at the same scanner could therefore help accounting for off-centre variations within scanner. Lastly, we did not monitor hydration state and time of day at scanning, factors which could also affect measures of brain microstructure [73].

Nevertheless, this work excels with its large number of participants and longitudinal design with closely timed acquisitions, rendering true effects of seasonal or age-related changes practically unlikely. Including two MRI systems which are usually linked by an upgrade, namely 3T Siemens Magnetom Skyra^fit^ as upgraded version of Verio, the consequences of such scanner upgrades on DTI outcome measures can be directly inferred from our study. Such across-upgrade investigations have been to date very rare [34,36].

## 5. Conclusions

In summary, based on two widely used Siemens MRI systems of the same field strength and two established DWI acquisition sequences we demonstrate that the reproducibility of DTI outcome measures strongly depends on imaging site, software and brain region. This is an alarming finding considering the importance of replicability of MRI assessments in the clinical context and increasing availability and diverseness of research-oriented MRI assessments on a large scale. It also underlines the necessity to carefully document, correct and adjust for different modifications of imaging parameters and applied data analysis pipelines. If not controlled for, such variations lead to much larger sample sizes which compensate the loss of statistical power. Our findings further support the use of the Gibbs ringing correction tool from Kellner et al. [48], encourage to adhere to one imaging system, scanning protocol and preprocessing pipeline and to conscientiously document every change in the aforementioned steps. Moreover, physiological effects such as ageing reflected in the decrease of FA were found to be robust against scanner differences and may be traceable despite variation in DWI data collection and processing, however, by the cost of a reduced effect signal and regional specificity. Regarding clinical applications, the potential impact of these variations on pathological changes should be kept in mind when assessing DWI data. Future studies need to further develop novel strategies to harmonise data acquisition and retrospective correction of hardware- and software-introduced differences in common MRI outcome measures to augment neuroimaging data reliability and replicability.

## Figures and Tables

**Figure 1 jcm-10-04987-f001:**
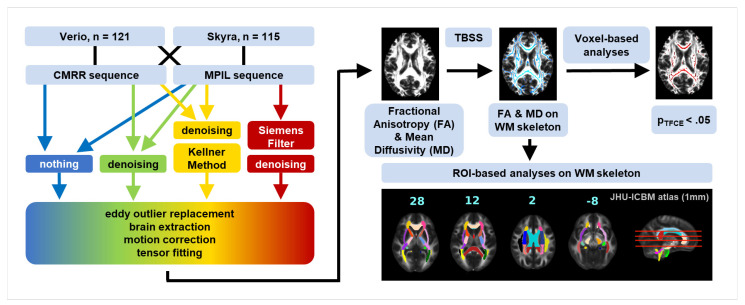
Analysis outline. DW images were collected at 3T Siemens Magnetom Verio and Skyra^fit^ with CMRR [57] and MPIL (Siemens product sequence) sequence respectively, resulting in four datasets. Each dataset from the MPIL sequence was processed with four different pipelines: no filtering (blue), denoising (green), denoising + unringing by “Kellner Method” (yellow) [48] and Siemens low-pass window filtering + denoising (red); datasets from the CMRR sequence were processed with three different pipelines (Siemens low-pass window filtering was not applied). Standard preprocessing steps followed these pre-filtering steps. By tensor fitting obtained FA and MD maps were skeletonised with tract-based spatial statistics (TBSS) [42] and fed into voxel-wise as well as ROI-based analyses on white matter (WM) skeleton.

**Figure 2 jcm-10-04987-f002:**
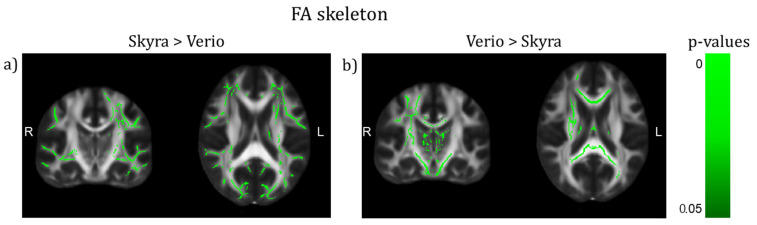
TBSS on the FA skeleton of scanner differences (CMRR sequence) after preprocessing with denoising (TFCE corrected, highlighted tracts: *p* < 0.05, (y,z) = (−18,19)). (**a**) More superficial WM tracts show higher values in Skyra than in Verio. (**b**) Rather deep WM tracts show higher values in Verio than in Skyra.

**Figure 3 jcm-10-04987-f003:**
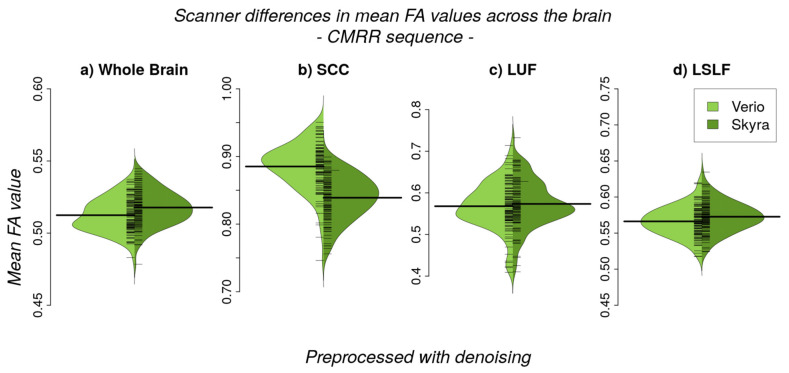
Scanner differences in the mean FA value of the white matter (WM) skeleton (CMRR sequence, after denoising) for (**a**) whole brain and region-of-interest analyses (**b**–**d**)). (**a**) On the whole brain level, Skyra delivers higher FA values than Verio (~1%, BF >> 3). (**b**) The splenium of the corpus callosum (SCC), (**c**) left uncinate fascicle (LUF) and (**d**) the left superior longitudinal fascicle (LSLF) highlight the differences in direction and magnitude of the scanner differences across ROIs: differences between scanners in percentages $\frac{mean\;FA\;value\;\left(Skyra\right)\;-\;mean\;FA\;value\;\left(Verio\right)}{mean\;FA\;value\;\left(Verio\right)}\;\times \;100$: SCC: ~−5.2% (BF >> 3), LUF: ~1% (BF = 1.1), LSLF: ~1.1% (BF >> 3).

**Figure 4 jcm-10-04987-f004:**
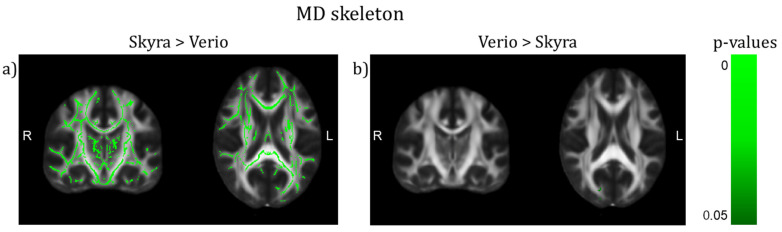
TBSS on the MD skeleton of scanner differences (CMRR sequence) after preprocessing with denoising (TFCE corrected, highlighted tracts: *p* < 0.05, (y,z) = (−18,19)). (**a**) The whole brain WM skeleton (except for a small part in the right occipital lobe) shows higher values in Skyra than in Verio. (**b**) Only a small white matter region in the right occipital lobe shows higher values in Verio than in Skyra.

**Figure 5 jcm-10-04987-f005:**
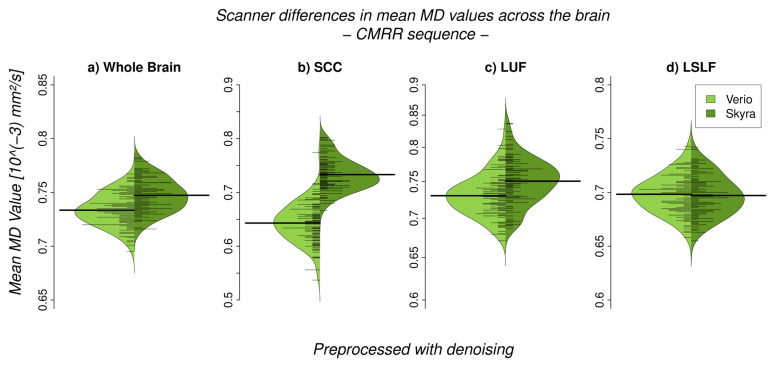
Scanner differences in the mean MD value of the white matter (WM) skeleton (CMRR sequence, after denoising) for (**a**) whole brain and region-of-interest analyses (**b**–**d**)). (**a**) On the whole brain level, Skyra delivers higher MD values than Verio (~2%, BF >> 3). (**b**) The splenium of the corpus callosum (SCC) and (**c**) the left uncinate fascicle (LUF) show as well higher MD values for Skyra (SCC: ~14% (BF >> 3), LUF: ~3% (BF >> 3)). (**d**) Only in the left superior longitudinal fascicle (LSLF) scanner differences are not pronounced: ~−0.2% (BF = 0.16). Percentages reflect relative differences between scanners: $\frac{mean\;MD\;value\;\left(Skyra\right)\;-\;mean\;MD\;value\;\left(Verio\right)}{mean\;MD\;value\;\left(Verio\right)}\;\times \;100$.

**Figure 6 jcm-10-04987-f006:**
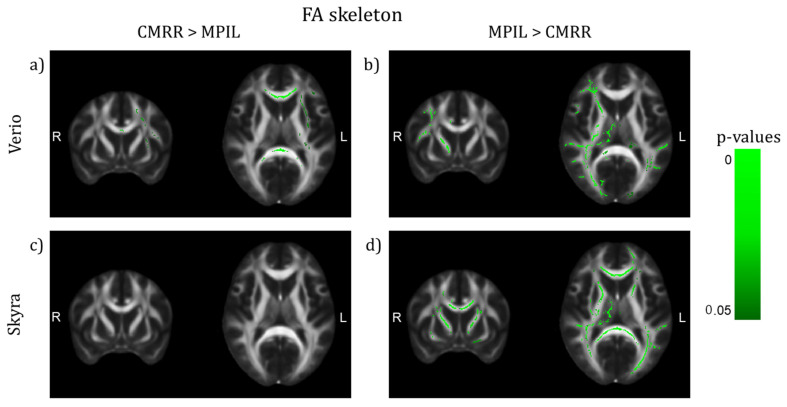
TBSS of differences on the FA skeleton between sequences after preprocessing with denoising (TFCE corrected, highlighted white matter areas: *p* < 0.05, (y,z) = (9,10)). (**a**,**b**) Verio: CMRR shows higher FA values in central brain areas, mainly in the CC, whereas MPIL shows higher FA values in lateral areas in the right hemisphere. (**c**,**d**) Skyra: MPIL delivers higher FA values in both hemispheres.

**Figure 7 jcm-10-04987-f007:**
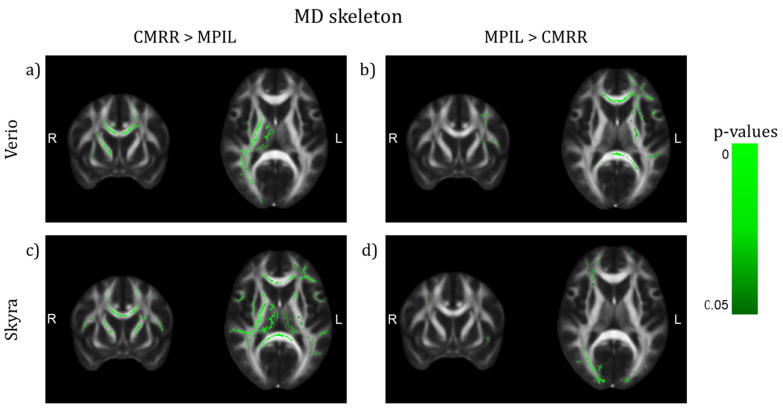
TBSS of differences on the MD skeleton between sequences after preprocessing with denoising (TFCE corrected, highlighted tracts: *p* < 0.05, (y,z) = (9,10)). (**a**,**c**) The CMRR sequence seems to deliver higher MD values in both hemispheres in central and lateral brain regions whereas the MPIL sequence shows higher MD values (**b**) in central and frontal regions and in the left hemisphere at Verio but (**d**) more occipitally and in the right hemisphere at Skyra.

**Figure 8 jcm-10-04987-f008:**
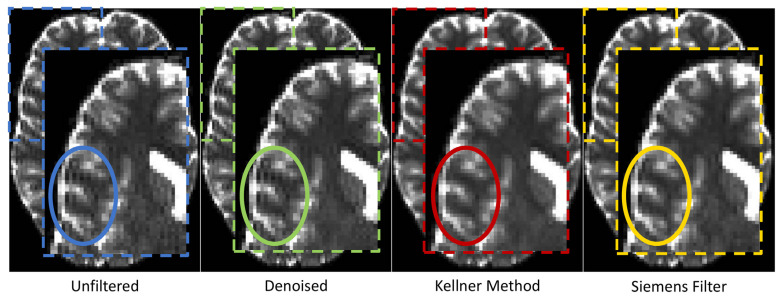
Appearance of Gibbs ringing artefact in b0 (T2-weighted) images after applying different preprocessing tools. Note that only the unringing tool (red) from Kellner et al. [48] considerably reduced GR as is most evident in the circled part of the b0 image.

**Figure 9 jcm-10-04987-f009:**
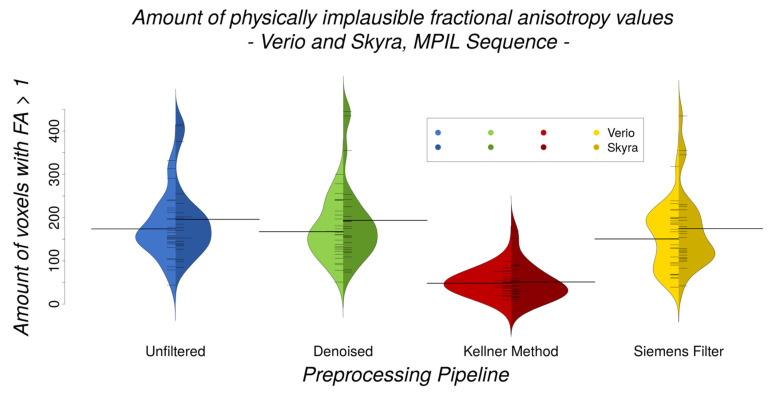
Amount of physically implausible FA values (Verio and Skyra, MPIL sequence, *n* = 23 + 28) is most reduced after unringing (red) with the Kellner Method (paired Bayesian *t*-test: BF[unfiltered ~ Kellner Method] > 8 × 10^9^) but does not differ significantly between scanners neither before nor after differing preprocessing pipelines (Bayesian linear modelling, full/null model comparison: $BF\left[\frac{\left(voxels\;with\;FA\;>\;1\right)∼scanner\;\times \;preprocessing\;pipeline}{\left(voxels\;with\;FA\;>\;1\right)∼preprocessing\;pipeline}\right]$ = 0.05 ± 1.09%).

**Figure 10 jcm-10-04987-f010:**
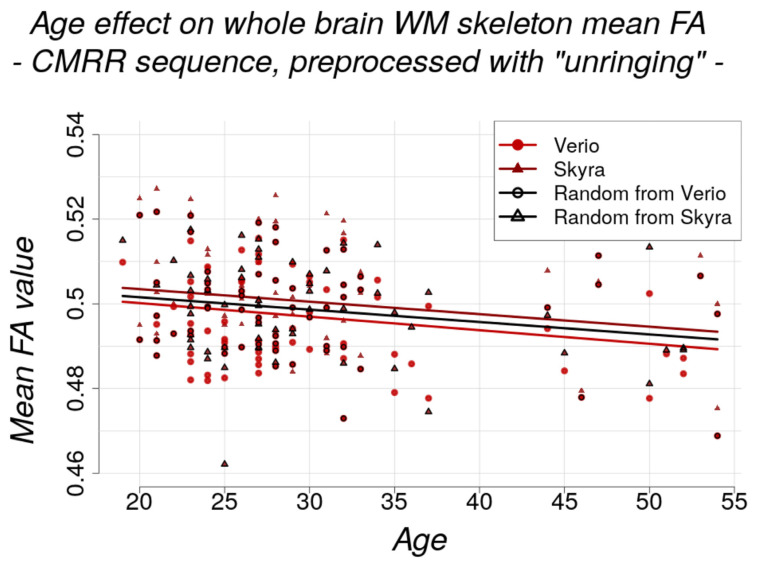
Negative effect of age on whole brain WM skeleton for both scanners separately (red circles: Verio, dark red triangles: Skyra, black framed circles and triangles: randomly selected to simulate pooled dataset from two different scanners). The effect modelled with linear modelling (mean FA value ~ age) is comparable between scanners (estimate ± std. error: Verio: Verio: −3.203 × 10^−4^ ± 1.237 × 10^−4^ and Skyra: −2.957 × 10^−4^ ± 1.347 × 10^−4^, random scanner: −2.925 × 10^−4^ ± 1.265 × 10^−4^). Bayesian linear modelling delivered significant results for the negative age effect: $BF\left[\frac{mean\;FA\;value∼age\;\times \;scanner}{mean\;FA\;value∼scanner}\right]$ = 5.49 ± 1.39%. In the case of studies conducted on different scanners (simulated by randomly selected subjects from Skyra and Verio, black line), the age effect size was still present and of intermediate magnitude.

**Figure 11 jcm-10-04987-f011:**
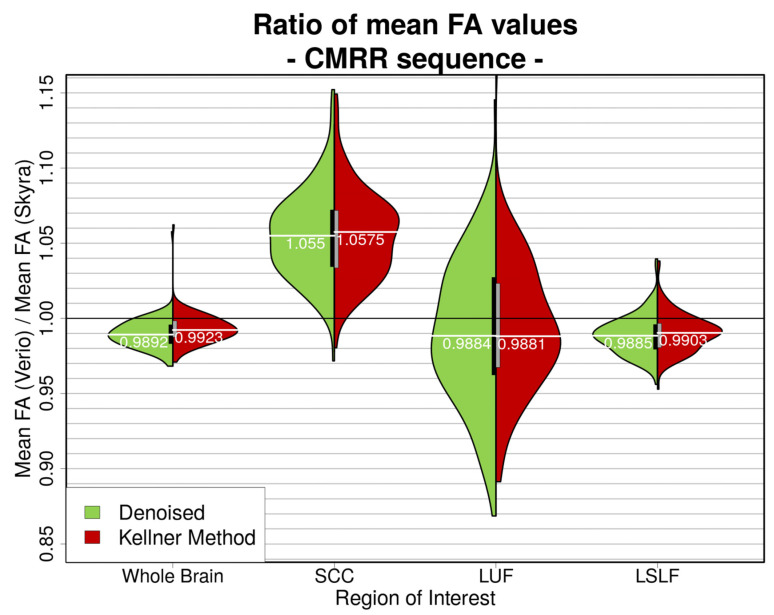
Ratio of whole brain mean FA values (Verio divided by Skyra) in different ROIs. For better visual clarity, unfiltered data is not shown. The left, green part of the violins represents denoised data; the right, red part shows the ratios after denoising and unringing. Bayesian linear modelling delivered significant differences between ROIs: $BF\left[\frac{mean\;FA\;ratio∼ROI\;\times \;preprocessing\;pipeline}{mean\;FA\;ratio∼preprocessing\;fpipeline}\right]$ >> 2 × 10196 ± 1.15%. Mean FA ratios calculated after different preprocessing pipelines do not differ significantly: BF[mean FA ratio ~ preprocessing pipeline] = 0.6.

**Table 1 jcm-10-04987-t001:** Mean FA values of the white matter skeleton at the whole-brain level and in different ROIs (CMRR sequence, after denoising). Bayesian linear modelling shows significant scanner differences for ROIs in the centre (SCC) and long white matter tracts (LSLF) but not for ROIs with curved lateral tracts (LUF).

WM Skeleton (After Denoising)	Scanner	Mean FA Value	SD	CoV |Verio—Skyra|(%)	Linear Model Bayes Factor (Mean FA Value ~ Scanner, *n* = 115)
whole brain	Verio	0.5124	0.0112	7.1	33.9
	Skyra	0.5177	0.0121		
SCC	Verio	0.7796	0.0168	30.7	1.1 × 10^32^
	Skyra	0.7631	0.0172		
LUF	Verio	0.5679	0.0566	31.5	1.1
	Skyra	0.5735	0.0571		
LSLF	Verio	0.5663	0.0204	11.1	3.3 × 10^12^
	Skyra	0.5727	0.0208		

**Table 2 jcm-10-04987-t002:** Mean MD values of the white matter skeleton at the whole-brain level and in different ROIs (CMRR sequence, after denoising). Bayesian linear modelling shows significant scanner differences on the whole brain level and for central and lateral ROIs (SCC and LUF) but scanner differences in the LSLF are not evident.

WM Skeleton (After Denoising)	Scanner	Mean MD Value [10^−3^ mm/s^2^]	SD [10^−3^ mm/s^2^]	CoV |Verio − Skyra| (%)	Linear Model Bayes Factor (Mean MD value ~ Scanner, *n* = 115)
whole brain	Verio	0.7336	0.0149	11.1	1.4 × 10^8^
	Skyra	0.7474	0.0155		
SCC	Verio	0.6431	0.0370	27.4	1.3 × 10^50^
	Skyra	0.7330	0.0289		
LUF	Verio	0.7310	0.0285	28.3	1.4 × 10^4^
	Skyra	0.7511	0.0321		
LSLF	Verio	0.6983	0.0173	13.9	0.16
	Skyra	0.6972	0.0186		

**Table 3 jcm-10-04987-t003:** Results of statistical analysis of head motion values (FD) with and without preprocessing with paired Bayesian *t*-tests. Denoising and unringing reduce head motion artefacts significantly. The CoVs of the preprocessing pipeline differences in head motion differ largely.

Contrast of Preprocessing Pipelines	Bayes Factor of Paired *t*-Test on Mean FD Values (*n* = 115)	CoV |Preprocessing Step − Preprocessing Step| (%)	
		Verio	Skyra
unfiltered ~ denoised	> 2 × 10^9^	20.5	27.4
unfiltered ~ Kellner Method	> 5 × 10^6^	27.8	40.2
denoised ~ Kellner Method	0.241	61.6	71.8

**Table 4 jcm-10-04987-t004:** Absolute head motion values per preprocessing pipeline and per scanner. Head motion estimated from the diffusion weighted images is significantly lower in Skyra than Verio independent of preprocessing pipeline (all BF >> 3). The CoV of the scanner difference in head motion is of comparable size between preprocessing pipelines.

Preprocessing Pipeline	Mean FD Value ± SD (mm)		CoV (%) |Verio − Skyra|
	Verio	Skyra	
unfiltered	0.417 ± 0.061	0.293 ± 0.064	41.9
denoised	0.355 ± 0.066	0.263 ± 0.066	47.3
Kellner Method	0.364 ± 0.067	0.269 ± 0.067	46.9

## Data Availability

Data (such as FA and FD values, ROIs, scanning protocols and data for age correlation) are stored at the Open Science Framework and openly available (https://osf.io/vnuqp/ (accessed on 5 July 2021)). Imaging data (anatomical and diffusion-weighted) are only available from subjects who signed additional consent (n = 57) and can be shared upon request (please contact witte@cbs.mpg.de).

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
