# Peer review of "Same Brain, Different Look?—The Impact of Scanner, Sequence and Preprocessing on Diffusion Imaging Outcome Parameters"

_jcm, 2021, doi:10.3390/jcm10214987_

Round 1
Reviewer 1 Report
I appreciate the effort put by the Authors in order to revise the manuscript, which results substantially improved. While the Authors have adequately addressed my previous concerns, I note the following remaining issues:
- “In addition, the gradient coils at the particular MRI scanners are regularly checked by service engineers from Siemens and by the “in-house” physicists. Further, daily phantom scans are run each morning before the first scan by the technical staff to ensure correct functionality of the hard- and software”. This description may appear rather generic. Given the importance of quality assurance in quantitative MRI, the Authors should describe in greater detail the quality controls they have performed, as well as any protocols they have adopted;
- “Concerning the quantification of the GR artefact, the linear least squares method employed by Lipsia to estimate the diffusion tensor in our preprocessing pipeline, also used in widely established DWI preprocessing software such as FSL’s dtifit, comes with the negative eigenvalue problem”. In DTI, given data heteroscedasticity, using the linear least squares method is conceptually inappropriate. The Authors should hence discuss and recognize this limitation;
- “In addition to the analysis of scanner differences in the FA skeleton for the different brain regions, we also tested for scanner differences in MD values (Figure 3 and 5). With TBSS, we found a clear whole brain difference with Skyra showing higher MD values than Verio (Figure 3; CMRR sequence; preprocessed with denoising)”. The Authors should discuss whether this finding (i.e. “this clear direction of the scanner difference”) is supported by phantom measurements and can reflect some miscalibration of diffusion weighting gradients;
- The Authors could consider the opportunity of discussing their findings with respect to the large multicentre and multiparametric phantom study by Fedeli et al (Phys Med 2021, 85: 98-106).
Reviewer 2 Report
The authors have responded to a number of the comments raised in the first review. A couple issues that were not addressed, however are
- It is unclear what is meant by the phrase “Bayesian statistics in a nutshell”
- The authors state no difference in image resolution existed between the two protocols. However, one used 6/8 and the other used 7/8 k-space coverage. This will influence image resolution regardless of the size of voxels used to digitize the image. Might this account for the differences observed in large vs. small white matter fiber tracts?
- Given the acknowledged limitations of the study, such as the potential confound of acquisition order (and factors such as reduced head motion in the 2nd scan hinting this could be a significant issue), inability to articulate differences between the CMRR and Siemens product pulse sequences, and the considerable reduction in the number of subjects described in Inter-sequence variability, the authors may want to temper some of the strongly-express conclusions. Specifically, the statement that the study finds “immense variations [due to instrumentation/study site] across ROIs in the WM skeleton’s mean FA and MD values have to our knowledge not yet been reported”, and extrapolation of the findings of this study to others and conclude it calls into “question recent attempts to harmonise multi-centre DWI data…” may be over-stating things.
Round 2
Reviewer 1 Report
Unfortunately, I am sorry to say that the Authors have not adequately addressed all my previous concerns. Therefore, I hope they are able and willing to solve these remaining issue in a further revision:
1) “In addition, the gradient coils at the particular MRI scanners are regularly checked by service engineers from Siemens and by the “in-house” physicists. Further, daily phantom scans are run each morning before the first scan by the technical staff to ensure correct functionality of the hard- and software”. This description may appear rather generic. Given the importance of quality assurance in quantitative MRI, the Authors should describe in greater detail the quality controls they have performed, as well as any protocols they have adopted.
Reply: We agree with the Reviewer that quality controls are of special importance in quantitative MRI. The daily quality control allows us to assure the correct functioning of all elements of the complex scanner hardware and software to exclude a random failure of e.g. coil elements which would affect the image quality. The gradient geometry and non-linearity are based on the hardware design and can not change significantly in a correctly working system. Therefore the regular check with the standardized phantom was chosen in our study to assure the accurate functioning of the system. In our study, we focus on diffusion-weighted imaging outcome measures which are calculated based on a single, not multiple b-values (b=1000 s/mm²), which is relatively low, and therefore the influence of the gradient non-linearities is also low and constant. We feel that elaborating more on quality assurance than our recommendation of quality assurance protocols for future studies as well as in clinical application would move away from the focus of our actual findings. We adapted the Discussion as follows (p.21): “Introducing quality control protocols and phantom scans to characterize scanner systems in single- and multi-site studies can also help to increase the reliability of diffusion-weighted measurements (as suggested by Fedeli et al., 2018). In the clinical context, we recommend to rescan a patient at the same MRI machine and to implement standardized quality controls”.
Given the importance of quality controls in quantitative diffusion-MRI, the Authors should describe in greater detail the “daily quality control” and “regular check with the standardized phantom” that they performed. Moreover, they should recognize that a lack of a specific check of diffusion gradients non-linearity is a limitation of this study.
2) “Concerning the quantification of the GR artefact, the linear least squares method employed by Lipsia to estimate the diffusion tensor in our preprocessing pipeline, also used in widely established DWI preprocessing software such as FSL’s dtifit, comes with the negative eigenvalue problem”. In DTI, given data heteroscedasticity, using the linear least squares method is conceptually inappropriate. The Authors should hence discuss and recognize this limitation.
Reply: We thank the Reviewer for questioning the linear least-squares method and allowing us to explain our choice: as we are only fitting tensors on DW images with a single b-value (b=1000 s/mm²) and the b0 images and not multiple diffusion weightings with different amounts of variance in function of the b-value, we do not have to deal with heteroscedasticity in fitting the data. Therefore, we are convinced that the linear least-squares method serves as an adequate tensor fitting procedure for this data with a good SNR and a relatively low b-value. In particular, the problem of the negative b-values would not change while employing e.g. a weighted least squares method, and a non-negative fitting method would not represent the acquired data appropriately. We hope that the Reviewer agrees that in this case, the linear least-squares method is appropriate and that the limitations we have already discussed regarding the LLS method (cf p. 23) are sufficient.
While the Authors have used only two b-values (i.e. 0 and 1000 s/mm2), I feel the use of linear least squares method still appears inappropriate, given data heteroscedasticity (Koay et al, J Magn Reson 2006, 182: 115-122). The Authors should hence recognize this limitation of their study.
3) “In addition to the analysis of scanner differences in the FA skeleton for the different brain regions, we also tested for scanner differences in MD values (Figure 3 and 5). With TBSS, we found a clear whole brain difference with Skyra showing higher MD values than Verio (Figure 3; CMRR sequence; preprocessed with denoising)”. The Authors should discuss whether this finding (i.e. “this clear direction of the scanner difference”) is supported by phantom measurements and can reflect some miscalibration of diffusion weighting gradients.
Reply: We are thankful that the Reviewer suggests this possibility of a miscalibration of the gradients to be the source of scanner differences. The locally varying directions of the scanner differences in MD measurements are a hint that a miscalibration is not the source of the differences as this would result in a global difference in the diffusivity measure. During the daily phantom scans, the gradient calibration is checked—as they are relevant for the diffusion weighting as well as for the signal encoding—and no deviations from normal functionality were detected.
As explicitly reported by the Authors, “with TBSS, we found a clear whole brain difference with Skyra showing higher MD values than Verio”. This “clear whole brain difference” could suggest a possible difference in calibration of diffusion weighting gradients between the two scanners. However, in their reply, the Authors submit that “the locally varying directions of the scanner differences in MD measurements are a hint that a miscalibration is not the source of the differences …”. This two affirmations appear as contradictory. Moreover, some small differences might be due to non-linearity effects of diffusion weighting gradients, that the Authors have not specifically assessed. The Authors should hence clarify in greater detail these issues.
4) The Authors could consider the opportunity of discussing their findings with respect to the large multicentre and multiparametric phantom study by Fedeli et al (Phys Med 2021, 85: 98-106).
Reply: We thank the Reviewer for the suggestion of discussing our findings with respect to the study by Fedeli et al. Fedeli and colleagues have furthered the field of quality assurance in quantitative diffusion MRI by setting up a QA protocol to be implemented in research as well as clinical applications that assesses the accuracy and spatial uniformity of estimated ADC values. Concerning our study though, we suppose that gradient non-linearities do not play a crucial role as explained in this part of the Discussion (p. 22):
“More detailed limitations include that we did not correct for gradient non-linearities which could similarly affect diffusion tensor metrics as the apparent diffusion coefficient as shown by Fedeli et al. (2018) and Tan et al. (2013) and thereby account for some of the differences we found on the mean FA and MD skeletons. However, the small non-linearity in the gradients of the used clinical MR systems, the relatively small field-of-view (only the brain), and the comparably low b-value (b=1000 s/mm²) minimize the effect of gradient non-linearity on image distortions and b-value variation compared to other studies where such corrections are needed (e.g. Human Connectome Project, van Essen et al., 2013, Jones 2010). To further control for a correct application of the diffusion gradients, the positioning of the participants followed a standardised protocol to position the brain in the isocenter of the gradient coil which presents the smallest non-liberalities. Additionally, in a parallel comparison of the anatomical images from Verio and Skyra, gradient non-linearity correction—conducted with the gradunwarp implementation [https://github.com/Washington-University/gradunwarp] in Python 2.7.)—did not substantially reduce the detected differences (Medawar et al., 2020). This is why we estimate that gradient non-linearities might only be a small share of the sources leading to the differences in the WM skeletons between the scanners”.
I agree with this additional discussion. However, the authors likely misunderstood my suggestion. Indeed, they have considered the study by Fedeli et al 2018 (Phys Med, 55: 135-141) instead of the study by Fedeli et al 2021 (Phys Med, 85: 98-106), which shows potential differences in quantitative ADC measurements due to different scanner characteristics. Please clarify and consider the opportunity of discussing this issue.
Author Response
Please see the attachment.

This manuscript is a resubmission of an earlier submission. The following is a list of the peer review reports and author responses from that submission.
Round 1
Reviewer 1 Report
This in vivo study aims to assess the impact of scanner, acquisition sequence and preprocessing on the estimate of brain DTI diffusion indices. The Authors have performed DTI measurements of 121 subjects, suggesting that DTI indices strongly depend on imaging site and software, with a bias varying across different brain regions. In general, the topic of the manuscript is of potential interest. However, the study presents a number of methodological concerns and various inaccuracies, which should be adequately addressed:
- The Authors have not performed/described any quality controls for assessing the correct application of diffusion weighting gradients. Nonetheless, in quantitative diffusion-MRI applications, it is fundamental to check the correct application of diffusion gradients (i.e. b-value) (Jones, Top Magn Reson Imaging 2010, 21: 87-99). Indeed, the proper application of diffusion weighting gradients is fundamental for guaranteeing accurate estimates of diffusion indices. The Authors should solve this important issue;
- The Authors did not apply any correction for non-linearity of diffusion gradients (e.g. Tan et al, J Magn Reson Imaging 2013, 38: 448-453), which has proven to be a potential non-negligible source of inaccuracy in quantitative diffusion-MRI of the body (e.g. Newitt et al, J Magn Reson Imaging 2015, 42: 908-919) and head (e.g. Fedeli et al, Phys Med 2018, 55: 135-141). The Authors should discuss this relevant inaccuracy of their study;
- As recognized also by the Authors, the lack of a test-retest analysis is a main limitation of this study. Indeed, the Authors cannot actually discriminate between inter-scanner and intra-scanner effects;
- In addition to the above limitations and issues, it should be noted that this study has considered only two scanners and acquisition sequences by the same manufacturer. Therefore, the general conclusions on effects of scanner and acquisition sequence submitted by the Authors might appear not necessarily warranted by results and rather speculative. The Authors should hence ponder the opportunity of revising the conclusions of the manuscript;
- The choice of approach/algorithm for estimating the diffusion tensor has been proven to bias the reliability of DTI-derived indices (Koay et al, Magn Reson Med 2006, 55: 930-936; Koay et al, IEEE Trans Med Imaging 2007, 26: 1017-1034). However, the Authors did not specify which algorithm they employed for estimating the diffusion tensor;
- The Authors have taken into account only the DTI-derived index of FA. It is not clear why the Authors have not considered also the diffusion index of MD, which can be straightforwardly derived from the same diffusion tensor estimated for obtaining FA. In order to improve their study, the Authors should perform their analysis for both FA and MD;
- Notwithstanding the numerous enrolled subjects, this study results of limited novelty, given various factors which include the low number of scanners/acquisition sequences and lack of MD analysis. An additional analysis of non-Gaussian diffusion indices (e.g. DKI-derived index of kurtosis) would have boosted the originality of this study.
Author Response
"Please see the attachment."

Reviewer 2 Report
This manuscript documents some of the effects of differences in data acquisition and analysis settings on diffusion MRI, with the objective of clarifying potential vulnerabilities in combining data from multiple scanners/sites in large-scale studies. Compared to previously published similar work, this study included a large number of participants. A number of factors are identified that can affect FA values, however the sources of these differences are not considered, and solutions (beyond the often unattainable recommendation of avoiding any differences between sites) are not provided. My specific comments are as follows:
- The analysis exclusively focuses on FA. Other diffusion MRI parameters (at minimum, mean diffusivity) could also be of interest.
- Differences between the Siemens product sequence and that of the CMRR sequence should be described, as it would help in the interpretation of observed differences.
- In Figures 2 and 4 there is a pattern where large/central WM bundles have one behavior, such as Verio > Skyra, and peripheral WM exhibits the opposite behavior. A possible reason for this behavior should be given. The implications for retrospective harmonization are appreciated, but without a sense of the source of the pattern, it is difficult to understand how these findings move the field forward. [Could these differences be attributed to differences in image resolution?]
- The lack of counterbalancing order of scanners between subjects is a substantial limitation. This should be specified in the methods as well as the discussion.
- The term “raw filter” on line 152 should be defined. It seems like an oxymoron.
- I was confused by the text in lines 226-228 – it seems not to say what happens if the Bayes factor is between 1/3 and 3. Also, I don’t agree that the sentence describes “Bayesian statistics in a nutshell”.
- Lines 238-239 seem to indicate more than half the subjects had to be removed from a comparison due to deviation from the “default parameters”. Given the considerable implication, a more complete description of this issue may be warranted.
- Regarding Gibbs ringing: The method for quantifying this phenomenon is not well motivated. For example, it seems Gibbs ringing could affect an FA measurement without causing it to exceed a value of 1. It also seems other artefacts could cause FA to be larger than 1. A more direct method of GR quantification is desirable. But also, this factor, like any post-acquisition step in the analysis, can be easily made uniform across sites in a multi-site study. Therefore, the contribution of the GR sections of this manuscript are unclear.
Author Response
"Please see the attachment."
